# UNC-45 assisted myosin folding depends on a conserved FX₃HY motif implicated in Freeman Sheldon Syndrome

Antonia Vogel[1,4], Renato Arnese [1,4], Ricardo M. Gudino Carrillo[1,2], Daria Sehr[1], Luiza Deszcz[1], Andrzej Bylicki[1], Anton Meinhart[1] & Tim Clausen [1,3] ✉

Myosin motors are critical for diverse motility functions, ranging from cytokinesis and endocytosis to muscle contraction. The UNC-45 chaperone controls myosin function mediating the folding, assembly, and degradation of the muscle protein. Here, we analyze the molecular mechanism of UNC-45 as a hub in myosin quality control. We show that UNC-45 forms discrete complexes with folded and unfolded myosin, forwarding them to downstream chaperones and E3 ligases. Structural analysis of a minimal chaperone:substrate complex reveals that UNC-45 binds to a conserved FX₃HY motif in the myosin motor domain. Disrupting the observed interface by mutagenesis prevents myosin maturation leading to protein aggregation in vivo. We also show that a mutation in the FX₃HY motif linked to the Freeman Sheldon Syndrome impairs UNC-45 assisted folding, reducing the level of functional myosin. These findings demonstrate that a faulty myosin quality control is a critical yet unexplored cause of human myopathies.

Muscle contraction relies on the sliding of actin and myosin filaments against each other. For efficient power generation, the two myofilaments are arranged in a quasi-crystalline manner in the sarcomere, the basic building block of all muscle cells[1]. Maintaining the higher-order organization of the sarcomere during exercise, proteotoxic stress, and ageing is of utmost importance, as reflected by the many myopathies connected with sarcomeric components[2]. To prevent damage and to oversee the highly specialized proteome of muscle cells, an extended set of molecular chaperones is employed[3]. Abundance and functional diversity of the involved folding factors reflect the intricate structures and assembly pathways of muscle proteins, such as myosin[3,4]. Notably, despite containing an ATPase domain that poses grand challenges for the cellular chaperone machinery[5,6], myosin is the most abundant protein in muscle cells, accounting for about 16% of the entire proteome[7].

Both the folding of myosin and the formation of thick filaments rely on the activity of multiple assembly factors[8], including members of the general chaperone machinery (e.g. HSP70, HSP90 and TRiC), but also muscle cell-specific chaperones. Among these components,

UNC-45 obtains a central role in coordinating myosin maturation[9–12]. In addition to functioning as a folding factor of the myosin motor, UNC-45 can form linear protein chains that provide regular spaced docking sites for HSP70 and HSP90. This molecular assembly line allows for coordinated chaperone activity on myosin heads protruding from the thick filament[13]. The important role of UNC-45 for myosin filament formation and sarcomere integrity is reflected by detrimental phenotypes upon genetic disruption. Knock-out of UNC-45 results in embryonic lethality, and both its upregulation and downregulation are linked to severe sarcomere defects[14,15]. Consistent with these findings, UNC-45 mutations have been directly linked to human myopathies[16,17].

UNC-45, which is present in all eukaryotes, employs a UCS domain to interact with myosin, a central domain that mediates oligomerization, and a TPR domain that interacts with members of the protein quality control (PQC) network[18]. The HSP70/HSP90 chaperones that bind to the TPR-domain are required for myosin assembly during muscle development[11,19]. In addition to coordinating filament formation, UNC-45 maintains the level of functional myosin in mature muscle cells during stress situations. To fulfil this role, UNC-45 uses its TPR

[1]Research Institute of Molecular Pathology, Vienna BioCenter, Vienna, Austria. [2]Medical University, Vienna, Austria. [3]Present address: Vienna BioCenter Core Facilities, Vienna, Austria. [4]These authors contributed equally: Antonia Vogel, Renato Arnese. ✉e-mail: tim.clausen@imp.ac.at

domain to interact with the ubiquitin ligase UFD-2, enabling the E3 enzyme to selectively target and ubiquitinate aberrant myosin molecules[20]. Consistent with this quality control function, stress situations applied to muscle tissues induce the interaction of UNC-45 with already assembled myosin molecules[21], whereas knockdown of UNC-45 results in an increase of myosin levels in the sarcomere[22]. However, the mechanism by which the chaperone distinguishes between folding competent and severely damaged myosin proteins and how it channels these into folding and degradation pathways is not understood. Moreover, and in contrast to the well-characterized general chaperones HSP70, HSP90, and TRIC[23–26], little is known about the substrate targeting mechanism of specialized folding factors such as UNC-45. Therefore, studying the interplay between UNC-45 and myosin is not only important to better understand how triage decisions are made in the PQC system but also to delineate the mode of action of client-specific chaperones.

By pursuing an integrative structural biology approach, we reveal that UNC-45 is able to bind myosin present in different conformations. Once bound to UNC-45, the fate of the myosin substrate depends on its folding state. We were able to delineate this mechanism by separating discrete UNC-45 complexes with "folding-competent" or "degradation-prone" myosin, and determining the high-resolution crystal structure of a minimal chaperone:substrate complex. We identified the conserved $FX_3HY$ motif in the myosin motor domain as primary site of UNC-45 interaction and applied cellular assays to demonstrate that this myosin epitope is essential for UNC-45 recruitment and maturation of the muscle protein. Strikingly, a Y582S mutation in this motif is linked to a severe developmental myopathy, the Freeman Sheldon Syndrome (FSS)[27,28]. The site-specific mutation abrogates the interaction between UNC-45 and myosin, causing severe protein misfolding in cells. These data directly relate myosin PQC to myosin-based myopathies, a connection that has not been studied in detail so far.

## Results

### UNC-45 forms separate complexes with folded and unfolded myosin

To characterize the interplay between chaperone and substrate at a molecular level, we reconstituted the complex between *C. elegans* UNC-45 and the motor domain of MHC-B muscle myosin (myo). After co-expression in insect cells, the UNC-45:myo complex was purified by tandem affinity chromatography and size exclusion chromatography (SEC)[20]. In the final SEC run, the UNC-45:myo complex eluted in a reproducible series of high- and low-molecular weight species (Fig. 1a). Since the stoichiometry of UNC-45 and myosin was similar in all fractions, we hypothesized that the elution peaks reflect different conformational states of myosin bound to UNC-45. To test this, we compared the SEC elution profiles of UNC-45:myo complexes with those of folded and heat-denatured myosin alone (Suppl. Fig. 1a). This comparison suggests that the high-molecular weight peak should represent UNC-45 bound to unfolded myosin (UNC-45:$myo^{UF}$), whereas the second peak, having an apparent molecular size of ~200 kDa, contains the chaperone engaged with a mostly folded substrate (UNC-45:$myo^F$). To explore the composition and dynamics of the distinct chaperone:substrate complexes, we subjected the two fractions to mass photometry[29]. The UNC-45:$myo^F$ sample displayed two discrete peaks, representing UNC-45 alone and a stoichiometric UNC-45:myosin complex. On the other hand, the UNC-45:$myo^{UF}$ sample displayed a polydisperse distribution of molecular weights ranging from 150–600 kDa, likely reflecting a heterogenous chaperone:substrate mixture with UNC-45 bound to misfolded and aggregated myosin molecules (Suppl. Fig. 1b). To further explore the folding state of the chaperone-bound myosin, we took advantage of the lower stability of myosin compared to UNC-45. Myosin, but not UNC-45, unfolds at 27 °C allowing us to selectively destabilize myosin at elevated temperature[30]. We thus incubated the two complexes at distinct

temperatures, either maintaining myosin stability (4 °C) or causing it to unfold (27 °C), and reapplied the samples to SEC analysis. For UNC-45:$myo^{UF}$, the elution volume remained almost unchanged after heat treatment, suggesting that the UNC-45-bound myosin is already unfolded (Fig. 1b). In contrast, when the UNC-45:$myo^F$ complex was incubated at 27 °C, the reapplied complex shifted to an earlier elution volume, characteristic of the unfolded myosin (Fig. 1c). These data strongly support our notion that myosin in the UNC-45:$myo^F$ complex is present in a mostly folded conformation, sensitive to heat-induced unfolding.

To identify interacting regions of chaperone and substrate, we performed crosslinking mass spectrometry (XL-MS) experiments. Analysis of the UNC-45:$myo^F$ complex containing the folded client protein showed that UNC-45 interlinks can be grouped into two classes (Fig. 1d, Suppl. Fig. 2a). They either comprise residues of intrinsically disordered UNC-45 regions (IDR class-I: loop 508–524 in central domain; UCS-loop 602–630; UCS C-terminus 931–961) or residues that are part of structurally well-defined sub-domains (class-II). While IDR class I contacts showed a promiscuous crosslinking pattern, the middle part of the UCS-domain (residues 695–720) is involved in specific class-II contacts mediated by its well-defined ARM-repeats. Crosslinked residues border the groove of the UCS domain that, based on structural alignment with β-catenin[13], has been suggested to form a myosin-binding canyon. We also observed specific class-II interlinks to the TPR domain of UNC-45, hinting to an additional myosin binding site. These data are consistent with an HSP70/HSP90 independent role of the TPR domain in yielding functional myosin[30]. Class-II crosslinks with myosin map to an N-terminal region (residues 30–150) adjacent to the SH3-like domain, residues 550–598 of the lower 50 kDa subdomain and the C-terminal part of the myosin motor adjacent to the converter region (Fig. 1d). Importantly, most interlinks are located on the same face of the myosin molecule, extending from the actin-binding site to the SH3-domain (Fig. 1e, Suppl. Fig. 2b). The defined crosslinking pattern between UNC-45 and MHC-B suggests that the two proteins bind to each other in a structurally defined manner as expected for folded proteins. Contrary, in the UNC-45:$myo^{UF}$ complex, UNC-45 is engaged in many unspecific interlinks with myosin (Fig. 1d). When plotting these interlinks, the unfolded nature of myosin is reflected by continuous streaks of dots (Suppl. Fig. 2a), suggesting that the relevant residues are part of disordered segments that arbitrarily interact with nearby sites. Moreover, the contacts are not restricted to clearly defined regions of myosin but rather involve all subdomains. To obtain further insights into the structural organization of the UNC-45:$myo^F$ complex, we used HADDOCK[31] to dock the myosin ATPase domain onto UNC-45, applying distance constraints provided by the XL-MS data. The protein-protein interaction with the best score had the myosin L50 domain bound to the UCS domain, close to the extended canyon implicated in myosin recognition (Suppl. Fig. 2b). In conclusion, the comparative SEC and XL-MS analysis indicate that UNC-45 is capable of binding its substrate in folded and unfolded conformations, yielding functionally different chaperone:substrate complexes.

### The folding state of myosin determines its interactions with PQC factors

To investigate whether the folding state of the captured substrate influences the channelling into folding and degradation pathways, we tested how the UNC-45:myo complexes interact with the molecular chaperone HSP90 and the UFD-2 ubiquitin ligase, respectively. Owing to the binding of HSP90 to the UNC-45 TPR domain[30], we performed interaction assays with the K82E mutant of UNC-45. As this mutant is known to abrogate TPR-dependent binding[13], it allows to selectively monitor the interaction of HSP90 chaperone with the captured myosin. Upon co-expression in insect cells, UNC-45$^{K82E}$:myo complexes isolated by tandem affinity chromatography showed a similar SEC

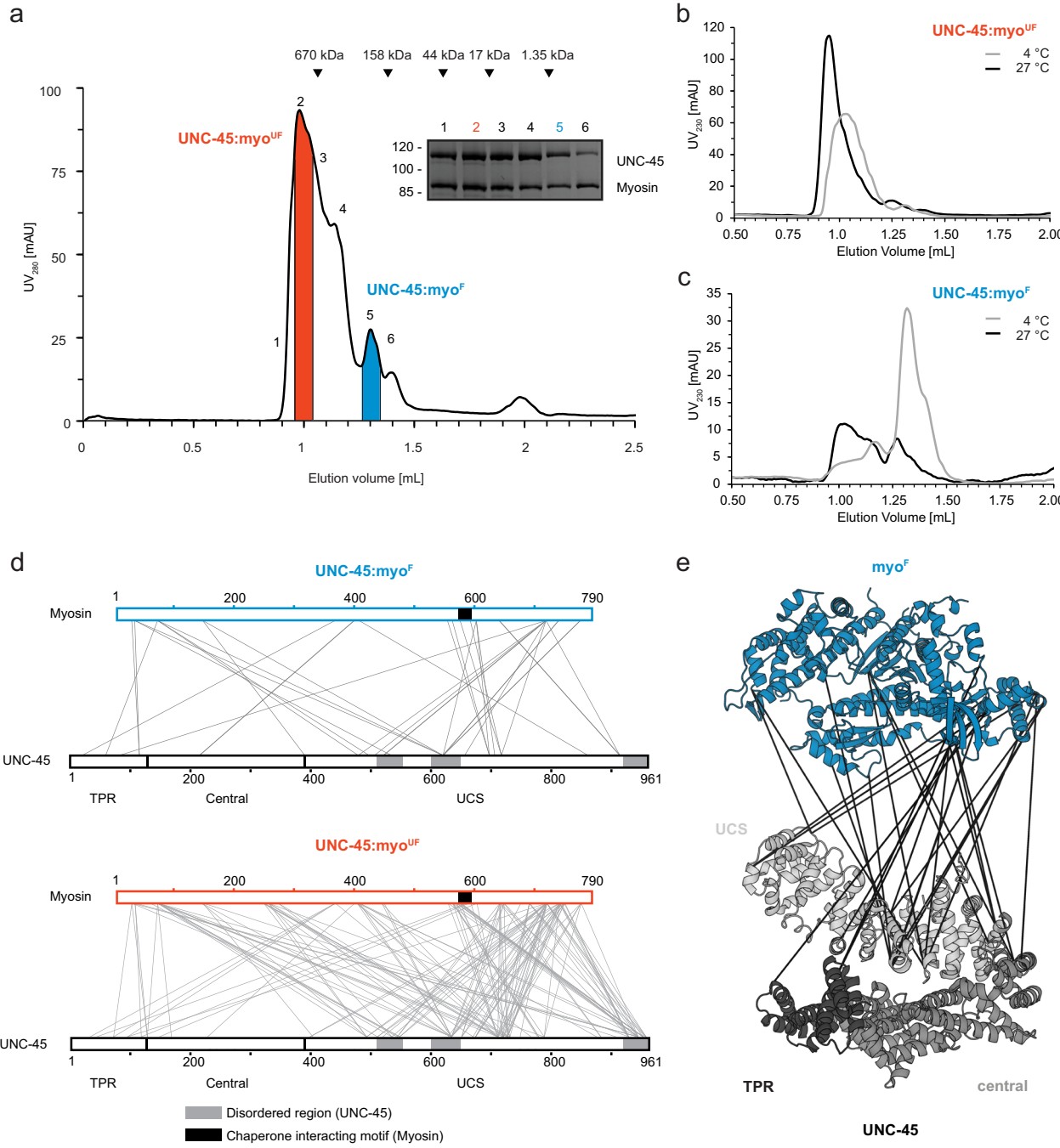

**Fig. 1 | UNC-45 can form distinct complexes with folded and unfolded myosin.**
**a** UNC-45:myosin complex purification, involving tandem-affinity chromatography and SEC. In the final SEC run, UNC-45 complexes with folded (myo$^F$) and unfolded (myo$^{UF}$) myosin can be separated. As seen in SDS-PAGE, chaperone and substrate associate in stoichiometric manner. For follow-up experiments, only the red (UNC-45:myo$^{UF}$) and blue (UNC-45:myo$^F$) coloured fractions were used. Molecular weight markers are indicated by arrowheads (670 kDa, 158 kDa, 44 kDa, 17 kDa, 1.35 kDa). **b**, **c** SEC analysis of UNC-45:myosin complexes incubated at 4 °C (grey) and 27 °C (black), with the higher temperature selectively destabilizing folded myosin. **d** XL-MS analysis of UNC-45:myosin complexes, showing interlinks with myo$^F$ (top) and myo$^{UF}$ (bottom). Illustrated crosslinks here and in the following figures have been filtered to FDR ≤ 5%. **e** Interlinks of the UNC-45:myo$^F$ complex are mapped on the crystal structures of UNC-45 (PDB: 4i2z, grey) and myosin (PDB: 6qdj, blue), highlighting the defined arrangement of the two proteins. Source data are provided as a Source Data file.

profile as the complexes with wildtype UNC-45, being engaged with myo$^F$ and myo$^{UF}$ (Suppl. Fig. 3a). In pull-down assays, HSP90 bound to both UNC-45$^{K82E}$ complexes containing either intact or unfolded myosin (Fig. 2a). This finding is consistent with the dual role of HSP90 in chaperoning folded and assembled myosin molecules in muscle sarcomeres[21] and serving as an E3 ligase co-factor in protein degradation pathways.

We next addressed the interplay between UNC-45, myosin and UFD-2, the ubiquitin ligase implicated in myosin quality control[20]. To map interaction sites between substrate, adaptor and E3 ligase, we applied an XL-MS approach. The three proteins were mixed in stoichiometric amounts and incubated at 4 °C or 27 °C, with the higher temperature selectively destabilizing myosin. Only when incubated at 27 °C, a complex between UNC-45, myo$^{UF}$ and UFD-2 was formed

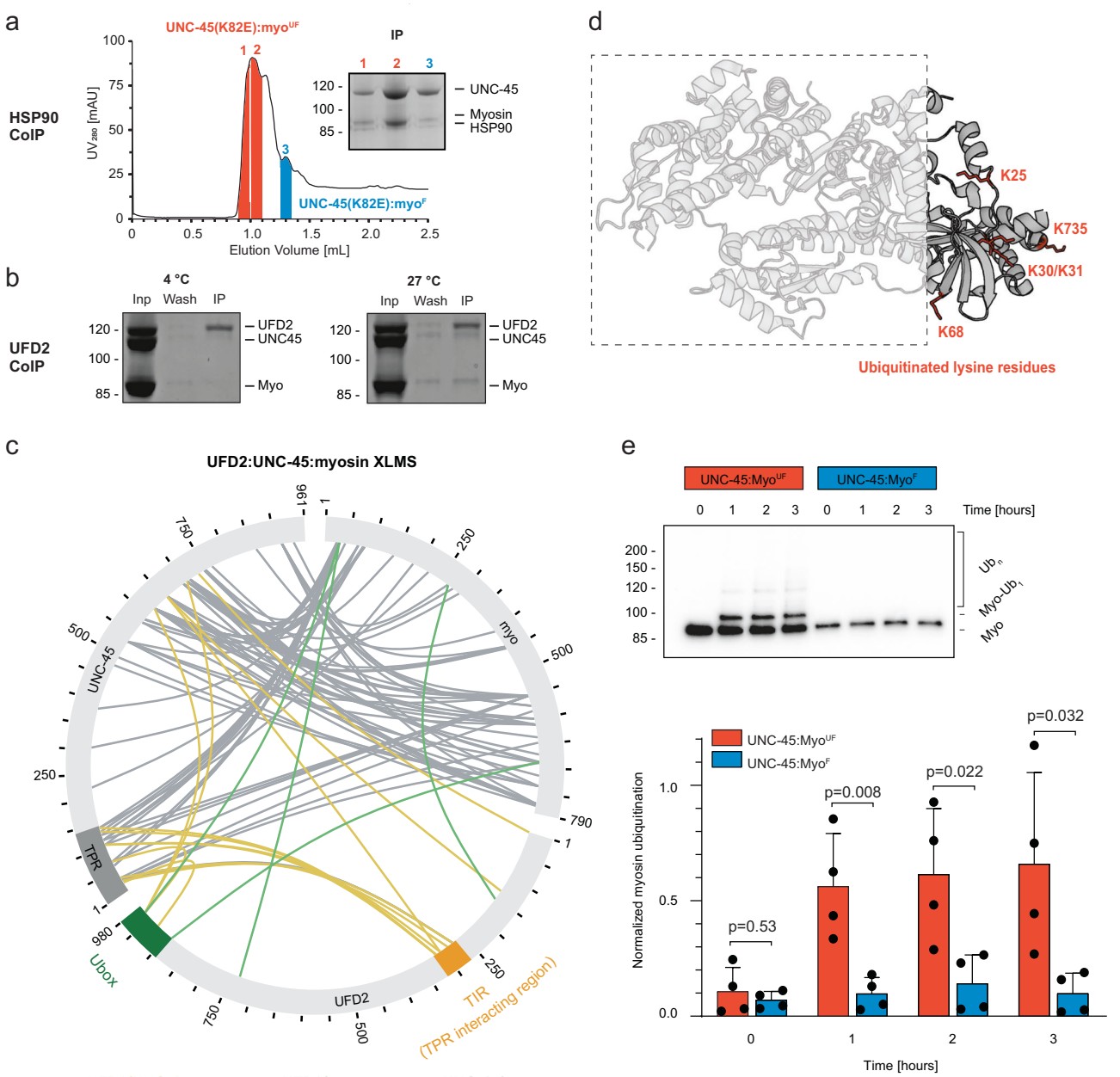

**Fig. 2 | PQC factors interact in a specific manner with UNC-45:myo^F and UNC-45:myo^UF complexes. a** Purification of UNC-45(K82E):myo complexes and pull-down with HSP90 probing TPR-independent interactions with recombinant HSP90, reveals binding to both myo^UF and myo^F. **b** Pull-down assay with UFD-2 reveals selective binding to UNC-45:myosin at 27 °C when myosin becomes unfolded. **c** XL-MS analysis of in vitro reconstituted UFD-2:UNC-45:myo^UF complex. Crosslinks between UFD-2/UNC-45 are coloured yellow; between UFD-2/myo^UF green and between UNC-45/myo^UF grey. **d** Mapping of UFD-2 mediated ubiquitination sites on myo^UF. Identified sites cluster in N- and C-terminal myosin segments near the

identified crosslinks to the catalytic Ubox domain of UFD-2. **e** UFD-2 mediated ubiquitination of natively purified UNC-45:myosin complexes, containing the folded and unfolded substrate, respectively. Shown is the anti-His Western Blot to selectively monitor myosin in UNC-45:myo^UF and UNC-45:myo^F complexes (top panel) and the quantification of the ubiquitin signal of myo^UF and myo^F (bottom panel). Data are presented as mean values and standard deviation. Two-tailed *t*-test was performed to assess statistical significance, with the *p*-values being indicated in the figure. *N* = 4. Source data are provided as a Source Data file.

(Fig. 2b), Recently it has been proposed that UFD-2 can interact with the TPR domain of CHN-1 and stimulate its E3 ligase activity through an internal EEYD motif near the catalytic Ubox domain[32] (Suppl. Fig. 3b). Our XL-MS data of the UFD-2/UNC-45/myosin complex indicates that UFD-2 also interacts with the UNC-45 TPR domain, however rather through an uncharacterized motif in the center of the ubiquitin ligase (TIR, residues 271–322), pointing to a different binding mode (Fig. 2c, Suppl. Fig. 3b). Interestingly, UFD-2 engages the myosin substrate in a specific manner, as evidenced by crosslinks between its catalytic Ubox

module and the terminal segments of the myosin motor domain (Fig. 2c). To test the functional relevance of the inter-crosslinks, we performed in vitro ubiquitination assays and mapped the ubiquitination sites on myosin. We observed that at higher temperature, when myosin is destabilized and complex formation is favoured, the ubiquitination reaction was markedly enhanced (Suppl. Fig. 3c), pointing to the selective targeting of misfolded myosin. Mapping the ubiquitination sites of myo^UF revealed that UFD-2 preferentially targets lysine residues in the N- and C-terminal regions of the motor domain, as

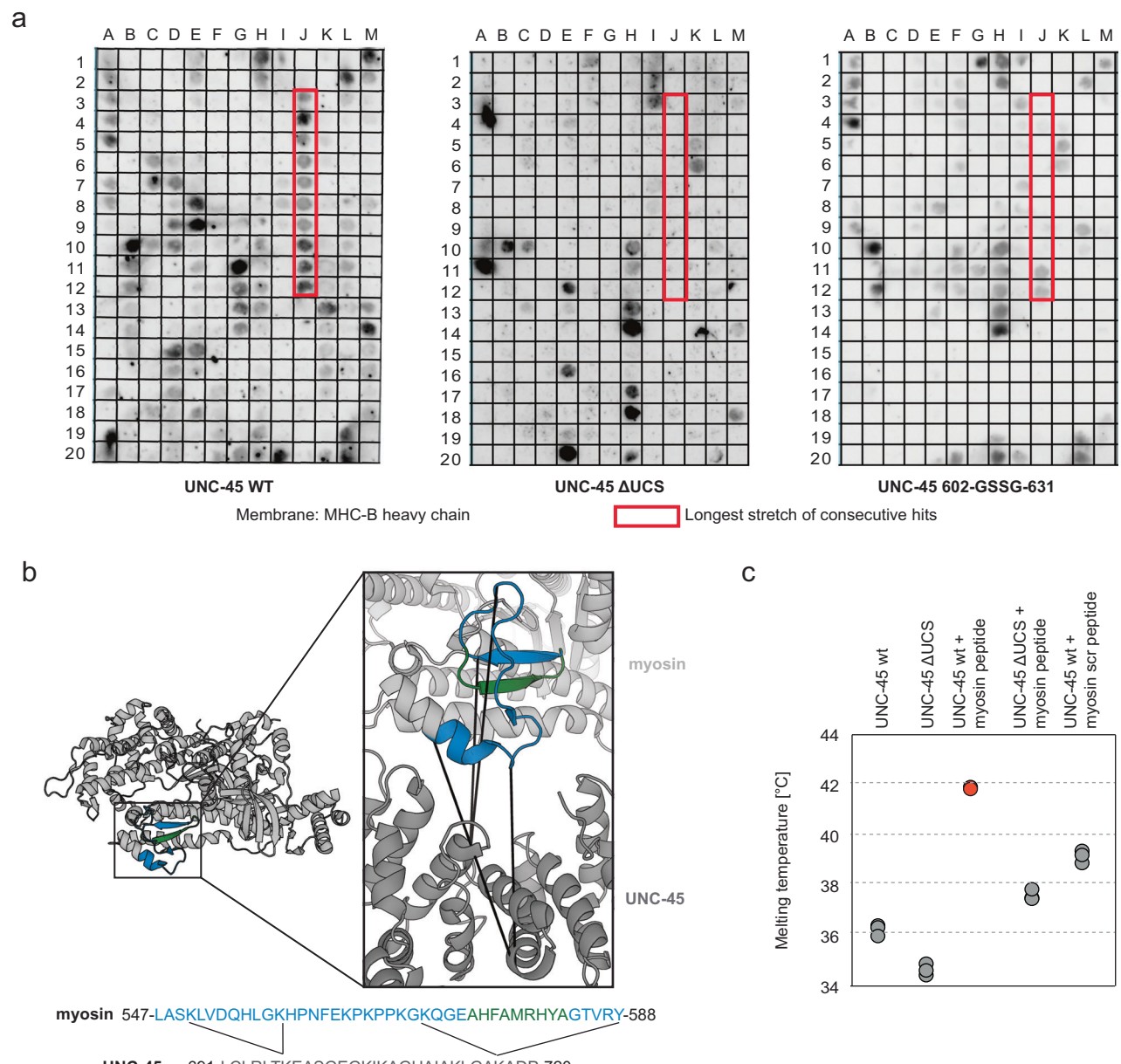

**Fig. 3 | UNC-45 recognizes a concise motif in the myosin motor domain.**
**a** Peptide spot analysis (MHC-B peptides of 15-residue per spot, with 12 residue overlap) of UNC-45 binding to myosin. ΔUCS and Δloop variants were used as binding-deficient controls. The longest stretch of consecutive hits is highlighted with a red box. **b** Mapping of the binding epitope on MHC-B myosin crystal structure (PDB: 6qdj). The inset highlights the consistency of the peptide spot data (highlighted in blue) with the XL-MS data (specific crosslinks undergone between myosin and UCS domain of UNC-45). The corresponding protein sequences are indicated below, together with the crosslinked residue and the later identified minimal binding motif (green). **c** Thermal shift assays addressing the binding of the identified motif to UNC-45. The ΔUCS chaperone and a scrambled (scr) myosin peptide were used as control. Assays have been performed in triplicates. Source data are provided as a Source Data file.

predicted by the XL-MS analysis (Fig. 2c, d). We thus propose that the central portion of the myosin motor, recognized and bound by UNC-45, is present in a mostly unfolded conformation enabling the N- and C-terminal segments to access the Ubox of the E3 enzyme. To further evidence the preference of UFD-2 for targeting unfolded myosin, we performed in vitro ubiquitination assays with natively purified chaperone:substrate complexes. We assessed myosin ubiquitination over time by Western Blot (Fig. 2e). We observed that UFD-2 exhibited a clear preference for ubiquitinating misfolded myosin contained in the UNC-45:myo$^{UF}$ complex, whereas the folded form present in the UNC-45:myo$^F$ complex was not targeted. In sum, our data show that UNC-45 can form distinct complexes with folded and unfolded myosin, which are then evaluated for structural integrity by the downstream acting

ubiquitin ligase UFD-2, leading to selective ubiquitination of aberrant client proteins.

## UNC-45 recognizes a distinct β-hairpin in the myosin motor domain

To better understand the structural requirements for UNC-45:myo complex formation, we analyzed whether UNC-45 recognizes a specific consensus motif. To this end, we synthesized a peptide library representing the motor domain of MHC-B and assayed the interaction with UNC-45. The longest stretch of consecutive peptides bound by UNC-45 comprised myosin residues 547–588 (Fig. 3a). To validate the interaction patch, we probed the membrane with two binding-deficient UNC-45 mutants, lacking either the entire UCS-domain (ΔUCS, lacking

residues 522–914) or the UCS-loop implicated in myosin binding[13,33] (Δloop, lacking residues 602–630). For both ΔUCS and Δloop variants, we observed several non-specific hits, however, they did not bind to the 547–588 motif recognized by wildtype UNC-45 (Fig. 3a). Notably, the 547–588 stretch corresponds to a β-hairpin in the motor domain. This motif is engaged in most class-II interlinks with UNC-45 in the XLMS datasets (Figs. 1d, 3b) and is located near the UCS domain in the molecular model obtained with HADDOCK (Suppl. Fig. 2b).

To confirm that the identified myosin stretch mediates UNC-45 binding in a specific manner, we characterized this interaction by thermal shift assay. In this approach, the stability of a protein – reflected by the melting temperature $T_m$ – is measured in the presence of ligands, assuming that ligand binding increases protein stability. To control for non-specific binding events, we used the UNC-45 ΔUCS mutant and a peptide with a randomly scrambled sequence of the myosin motif (Fig. 3c, Suppl. Fig. 4a). Upon addition of the 547–588 myosin peptide, the melting temperature increased by 5.5 °C, pointing to a direct interaction with UNC-45. Consistently, the ΔUCS mutant was not stabilized by the myosin peptide. Likewise, the scrambled peptide failed to stabilize wildtype UNC-45 to the same extent as the myosin peptide. Fluorescence anisotropy measurements, which were carried out in a qualitative manner due to low affinity of the binder, confirmed the specific interaction between the identified myosin ligand and UNC-45 (Suppl. Fig. 4b). In sum, these data indicate that residues 547–588 of the myosin motor domain contain the primary interaction site for UNC-45.

## Structure of minimal UNC-45:myosin complex reveals FX₃HY motif

To investigate the structural basis of myosin recognition, we pursued a crystallographic approach. Given the difficulties in obtaining a co-crystal structure with *C. elegans* UNC-45, we moved to the less complex fungal orthologue SHE4. Despite lacking the TPR-domain, SHE4 contains a functional UCS-domain for myosin binding[13,34]. Moreover, sequence alignments indicated that the myosin motif recognized by UNC-45 is conserved in fungal isoforms (Fig. 4a) and has been previously postulated to be part of the SHE4 binding motif[34]. Bioinformatic analysis revealed that *Kluyveromyces lactis* SHE4 is the most promising candidate for crystallographic studies, having a functional UCS domain but lacking various flexible, non-conserved regions that impeded crystallization of the yeast ortholog[34] (Suppl. Fig. 5a). In fact, the 76 kDa protein readily crystallized, enabling us to determine the structure of wildtype SHE4$_{KL}$ at 2.4 Å resolution (Suppl. Fig. 5b, Supplementary Table 1). Overall, SHE4$_{KL}$ resembles an L-shaped superhelix that is composed of a short, flattened N-terminal arm and an extended, slightly bent UCS-domain that aligns well with the UCS-domain of *C. elegans* UNC-45 (root mean square deviation: 3.8 Å for 321 Cα-atoms) (Suppl. Fig. 5b, c). Both fungal and metazoan UCS-domains are composed of a series of Armadillo repeat motifs (ARM), each of which is defined by a short helix (H1) followed by two elongated helices (H2, H3) that stack in a perpendicular manner (Suppl. Fig. 5b). In the resulting superhelical UCS-scaffold, the ARM-repeat H3 helices line the concave face of the substrate-binding canyon to mediate interactions with myosin[35].

To characterize the structural motif recognized by the UCS domain, we synthesized a series of Myo4 peptides, containing the β-hairpin motif 547–588 identified in the peptide spot assay. Isothermal titration calorimetry (ITC) measurements showed that 13 residues (561–573) comprise the core of the myosin motif, binding to SHE4$_{KL}$ with a $K_D$ of 0.6 μM (Fig. 4b). After intense co-crystallization trials with various myosin peptides, we obtained a well-diffracting SHE4 co-crystal with a peptide comprising Myo4 residues 561–581. We determined the co-crystal structure at a resolution of 2.4 Å and were able to visualize a minimal SHE4:Myo4 complex in atomic detail. In this structure, the Myo4 residues Lys561-Phe562-Ile563-Val564-Ser565-

His566-Tyr567 were well-defined by electron density (Fig. 4e). The eight myosin residues are accommodated in the UCS-canyon by a dense network of hydrogen bonds and hydrophobic interactions. The tight binding mode forces the Myo4 peptide to adopt a bent conformation, with its C-terminal portion being directed out of the central groove of the UCS-domain. Notably, this bent conformation is distinct from the β-turn observed in the native myosin structure, preventing UNC-45 to target functional myosin (Suppl. Fig. 6). As for its substrate, the interacting residues in the UCS domain of SHE4 are mostly conserved across species (Suppl. Fig. 7) suggesting that the observed binding mode is generally relevant. To validate the structural data, we mutated the strictly conserved Asn475 that is located in the UCS binding groove and binds to the backbone of the captured myosin peptide (Fig. 4e, Suppl. Fig. 7). ITC-measurements showed that binding to the myosin peptide was abrogated in the N475A mutant, confirming its participation in substrate binding (Fig. 4c).

Given the conservation of the myosin residue Phe577, His581 and Tyr582, which are accommodated in specific pockets of the UCS canyon (Fig. 4a–d), we tested the general relevance of these residues in the *C. elegans* system. Since UNC-45 precipitated during ITC measurements, we performed thermal shift measurements, incubating UNC-45 with wildtype and mutant MHC-B peptides. Contrary to the native peptide that elevated UNC-45 stability by 5.5 °C, introducing the F577A, H581A or Y582A mutations only resulted in a minor increase in $T_m$ (Suppl. Fig. 8a). Consistently, fluorescence anisotropy measurements supported the crucial role of these residues in mediating UNC-45 binding, as F577A, H581A and Y582A myosin peptides were strongly impaired in their interaction with UNC-45 (Fig. 5a). Furthermore, the binding of this motif to UNC-45 fits well to our crosslinking data (Fig. 1d), according to which myosin residues 576–587 are accommodated in the center of the UCS canyon, juxtaposing the adjacent motor domain regions to the UNC-45 chaperone. The observed crosslinks suggest that the complex is held together on one side by links from myosin residues Lys550 and Lys558, and on the other side by a link from residue Tyr588 (Fig. 3b). In concert, these interactions yield a chaperone-substrate complex with a well-defined topology that locks the β-hairpin motif in place. Taken together, our structural and biochemical data indicate that the conserved myosin residues Phe577, His581 and Tyr582 mediate binding to *C. elegans* UNC-45. We propose that the conserved FX₃HY motif represents a universal and highly specific substrate epitope, recognized by the myosin-specific chaperone UNC-45.

## Disruption of the FX₃HY motif hinders myosin folding in the cell

To test the functional relevance of the FX₃HY-motif in a cellular context we monitored UNC-45 dependent maturation of myosin in insect cells[30]. Since UNC-45 is essential to produce soluble myosin, UNC-45/myosin co-expression can be used as a proxy for evaluating chaperone-substrate interactions. By assessing the yields of soluble myosin produced in insect cells, we found that replacing the β-hairpin loop (581-HYAG-584) with a GSGS spacer impairs UNC-45-dependent myosin folding (Fig. 5b). Importantly, the same deleterious effect was observed for the single-site mutants F577A, H581A or Y582A (Fig. 5b, c), highlighting the impact of the FX₃HY motif on myosin maturation. We then looked for disease causing mutations in this motif and found that the Freeman-Sheldon Syndrome (FSS), a myopathy associated with severe multiple congenital contracture[28], is linked to the substitution of a serine at the conserved Tyr582 site. To explore whether the Y582S mutation may impact interaction with UNC-45, we co-expressed the respective MHC-B variant with UNC-45 in insect cells and observed that this mutation significantly decreased the levels of soluble myosin (Fig. 5c). Fluorescence anisotropy measurements with a myosin peptide carrying the Y582S mutation confirmed the in vivo data, as binding to UNC-45 was markedly reduced (Fig. 5a). The resultant misfolding of myosin may underlie the phenotype observed in a *Drosophila*

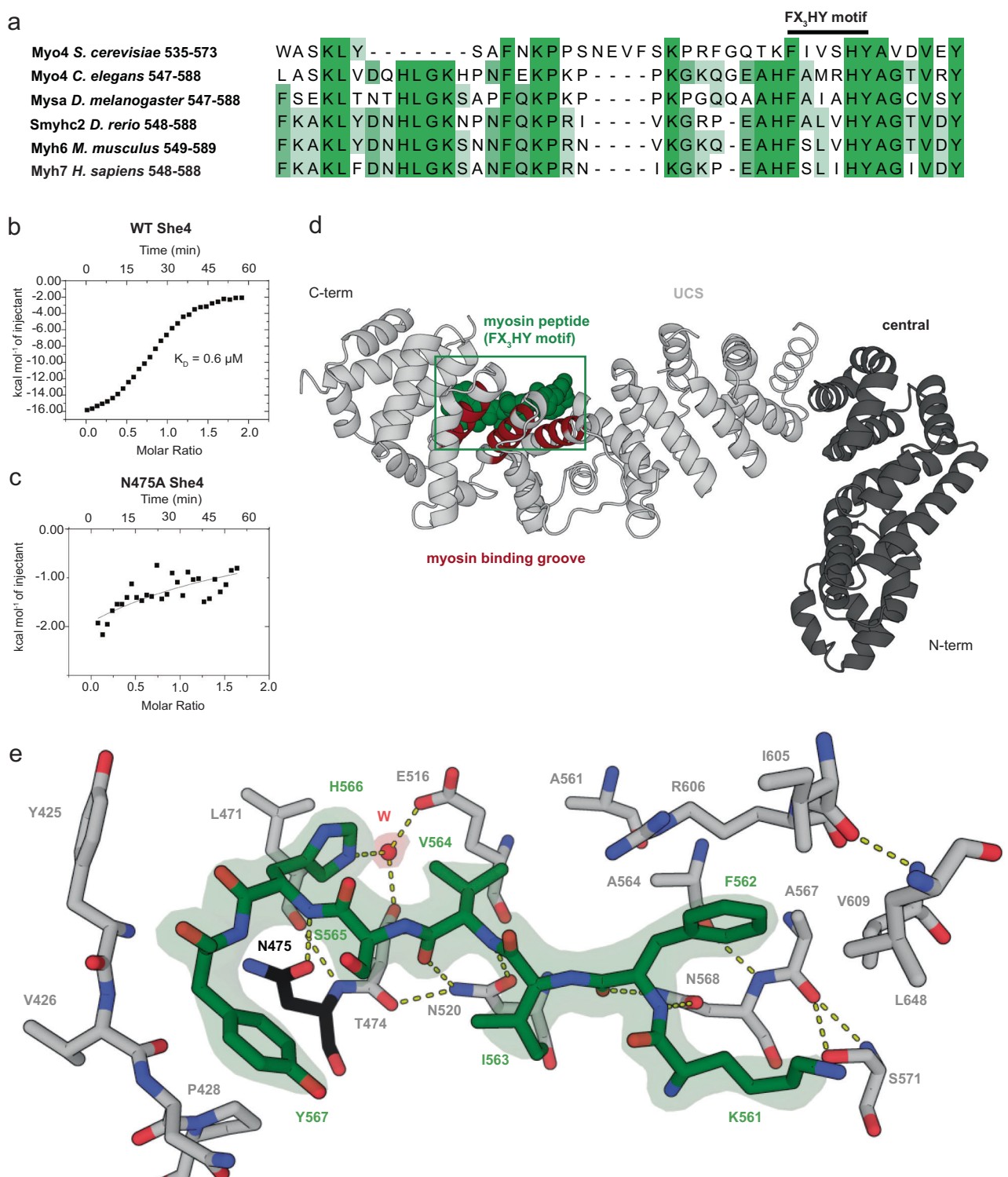

**Fig. 4 | Crystal structure of a minimal UNC-45:myosin complex. a** Sequence alignment of the identified UNC-45 interaction motif in MHC-B (Myo4_Cel). Residues are coloured according to sequence identity (dark green > 80%, white < 40%). Conserved residues that define the FX₃HY motif are labelled. **b** ITC measurement of Myo4 peptide (residues 561–573) binding to SHE4, the yeast ortholog of UNC-45. **c** As in (**b**) with N475A SHE4, a mutation in the UCS canyon that abolishes myosin binding. **d** Co-crystal structure of the minimal SHE4:Myo4 complex, highlighting the bound myosin peptide defined by the electron density (residues 561–567,

green) and the central part of the myosin binding groove (brown). UCS and central domain are depicted in grey and black, respectively. **e** Detailed view on SHE4:Myo4 interactions, with key residues being labelled and selected hydrogen bonds shown in yellow and a coordinating water molecule in red. Asn475 mutated to Ala in functional assays is coloured in black. The 2Fo-Fc electron density of the peptide is contoured at 1.5 $\sigma$ and is rendered as isodensity surface. Source data are provided as a Source Data file.

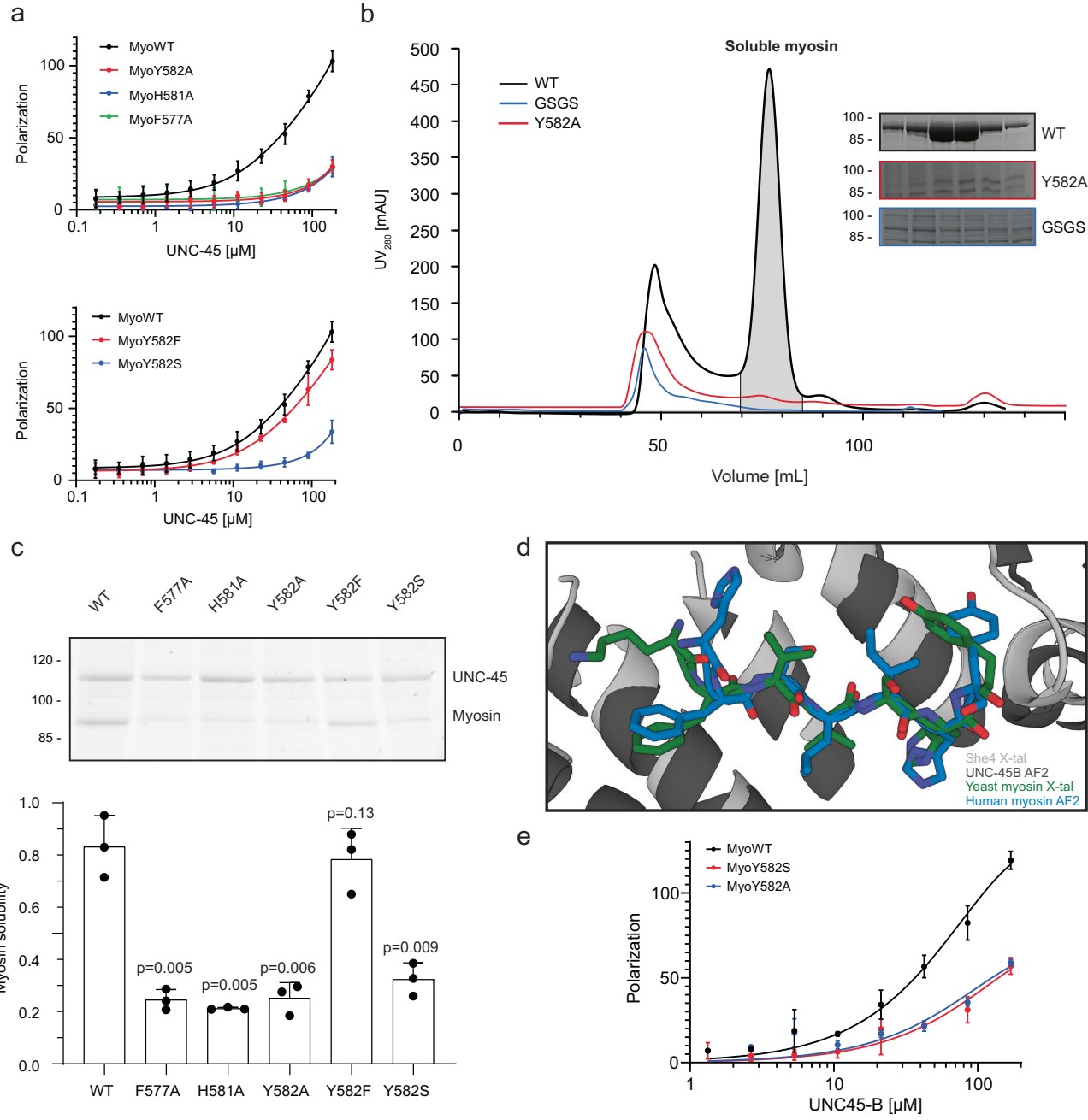

**Fig. 5 | The FX₃HY-motif is essential for UNC-45 binding and myosin folding.**
**a** Fluorescence polarization experiments comparing a wildtype myosin peptide with mutants of the conserved FX₃HY motif. Increasing concentrations of purified UNC-45 were titrated to fluoresceine labelled peptides. Data are presented as mean values and standard deviation. $N = 3$ (**b**) SEC analysis of purified myosin motor domain comparing wildtype, Y582A (red) and R580-GSGS-T585 (blue) variants. Fractions eluting at the size of the functional motor domain (~75 ml, grey peak) were analyzed by SDS-PAGE. **c** In cell solubility assay of myosin variants containing mutations in the FX₃HY motif. Cleared insect cell lysates were analyzed by SDS-PAGE and the relative amount of soluble myosin quantified by comparing protein

levels to UNC-45. Data are presented as mean values and standard deviation. Two-tailed $t$-test was performed to assess statistical significance, with the $p$-values being indicated in the figure. $N = 3$ (**d**) AlphaFold2 model of UNC45B (dark grey) in complex with MYH3 peptide (blue) aligned to the co-crystal structure of She4 (light grey) with the FX₃HY containing peptide (green). **e** Fluorescence polarization measurements comparing binding of a wildtype MYH3 peptide with Y582A and Y582S variants of the FX₃HY motif. Increasing concentrations of purified UNC45B were titrated to fluoresceine labelled peptides. Data are presented as mean values and standard deviation. $N = 3$. Source data are provided as a Source Data file.

*melanogaster* FSS disease model, in which the Y582S mutation led to a progressive disruption of muscle sarcomeres, although the kinetic properties of the isolated mutant were more similar to wildtype myosin when compared to other FSS mutants[27]. Contrary to the Y582S mutation, introducing a Y582F mutation had milder effects. In fluorescence anisotropy experiments, the Y582F mutant showed slightly lower UNC-45 affinity compared to the native peptide (Fig. 5a), while in

the cellular chaperone assay, the amounts of soluble myosin were comparable to that of wildtype protein (Fig. 5c). Consistent with our biochemical data, *C. elegans* non-muscle myosins contain a Phe582, suggesting that the phenyl ring at this position is sufficient for UNC-45 assisted folding (Suppl. Fig. 8b). With regards to the human chaperone UNC45B, the overall sequence identify is only 35%; however, residues in the UCS domain involved in FX₃HY binding are highly conserved

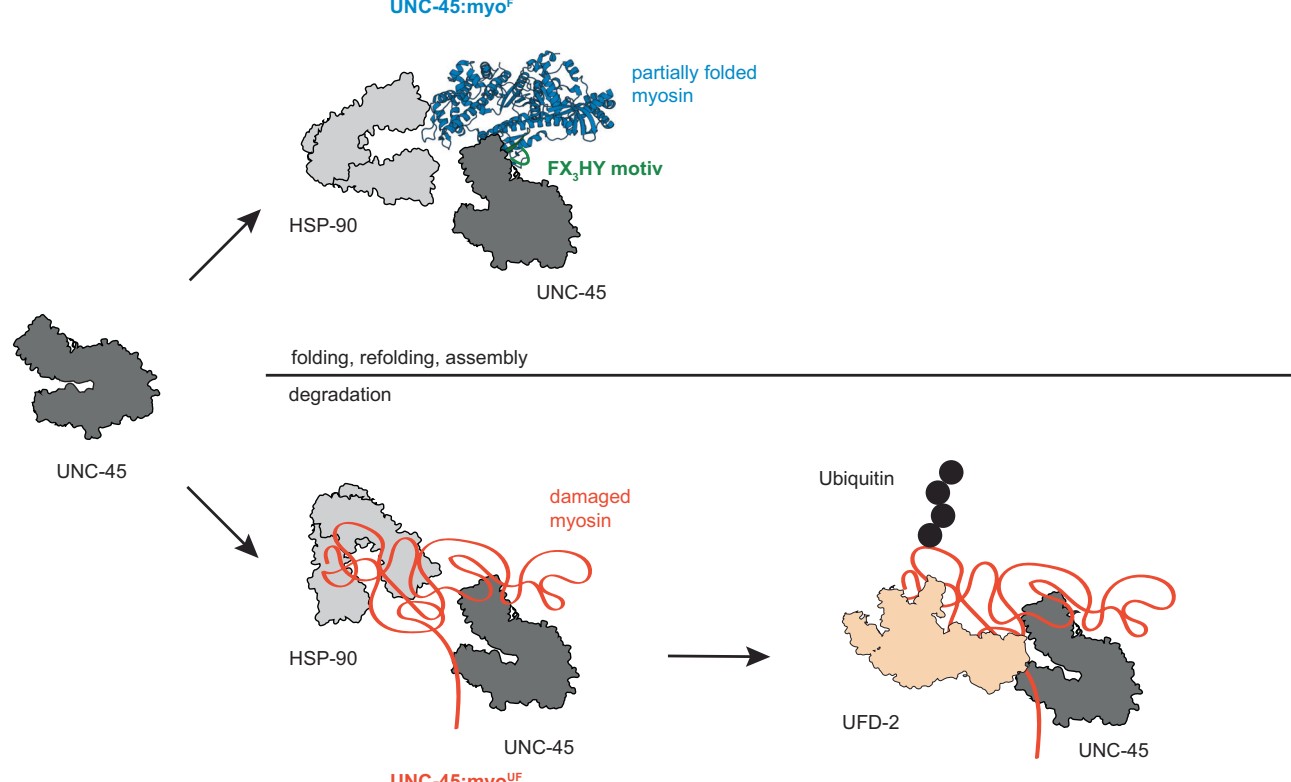

**Fig. 6 | Model of the interplay of UNC-45 with partner PQC factors to determine the fate of its myosin client.** UNC-45 (dark grey) can bind to myosin present in different conformations (folded or unfolded; recognized by the $FX_3HY$ motif) and channel it into folding or degradation pathways. In the UNC-45:myo$^F$ complex shown in the top panel, the myosin motor domain exists in a close to native conformation (blue), as expected for a late folding intermediate prior to incorporation into thick filaments or resulting from mild proteotoxic stress. HSP90 (light grey) is involved in this assembly pathway as well. Contrary, irreversibly damaged myosin (orange) captured in the UNC-45:myo$^{UF}$ complex is channelled by UNC-45 and its partner chaperone HSP90 to the quality-control ubiquitin ligase UFD-2 (light orange), inducing its poly-ubiquitination and proteasomal degradation.

(Suppl. Fig. 7). We thus tested to what extent the identified binding mode is generally relevant. We first used AlphaFold2[36] to predict the complex of human UNC45B with MYH3, the embryonic myosin isoform mutated in FSS[28]. However, the predicted complex did not fit to our XL-MS data, and the identified $FX_3HY$ motif was not engaged in complex formation, likely because the motif is buried in the folded motor domain. We therefore modelled a minimal complex comprising the UCS domain of UNC45B and the $FX_3HY$ motif of myosin. In this case, we observed an interaction that closely mimics the binding mode identified in our co-crystal structure (Fig. 5d, Suppl. Fig. 9b). To validate the prediction, we reconstituted the interaction of the human proteins in vitro, using recombinant human UNC45B (Suppl. Fig. 9c). We performed fluorescent anisotropy measurements with MYH3 derived peptides containing the $FX_3HY$ motif and observed chaperone:substrate interactions that phenocopied data of the *C. elegans* system: a weak but significant binding of the wild-type peptide that got strongly reduced by specific point mutations (Fig. 5e). Taken together, these data demonstrate the conserved role of the myosin $FX_3HY$ motif to mediate interactions with the cognate UNC-45 chaperone, as required for protein maturation.

## Discussion

Structural characterization of chaperone-substrate interactions is challenging due to the heterogeneity of these complexes, which contain a continuum of transient folding intermediates. Advances in cryo-electron microscopy and NMR spectroscopy allowed visualization of certain chaperone-substrate complexes, delineating the targeting mechanisms of general chaperones including HSP90, HSP70 and TriC[26,37–45]. Conversely, the mode of action of the many client-specific folding factors is less understood, with notable exceptions being the histone and Rubisco chaperones[46,47]. To better understand substrate targeting by specialized chaperones, we studied the interplay between UNC-45 and its client protein myosin. As AlphaFold2 failed to predict this interface—likely due to myosin present in a non-native conformation – we applied an integrative structural biology strategy, using protein crystallography and XL-MS as key methods. We show that the highly conserved $FX_3HY$ motif in the myosin motor domain is a discrete molecular epitope recognized by the UNC-45 chaperone, which then presents the bound client to downstream PQC factors.

An important feature of the PQC decision-making process is that UNC-45 can accommodate both folded and misfolded myosin. Upon forming separate complexes with the two myosin forms, downstream acting chaperones and E3 ligases channel the presented client into folding or degradation pathways (Fig. 6). Whereas the general chaperones, as shown here for HSP90, seem to obtain an indirect role preventing premature myosin aggregation and keeping the (partially) misfolded protein in solution, the UFD-2 ubiquitin ligase takes the decision. Only the severely damaged myosin is targeted for degradation whereas folded myosin in complex with UNC-45 is channelled into assembly pathways. Importantly, for myosin ubiquitination by UFD-2, formation of a productive E3 ubiquitin ligase complex depends on a composite signal, provided by the UNC-45 adaptor (available TPR docking site, not occupied by HSP70/HSP90 chaperones) and the captured myosin (misfolded state). Furthermore, we noted that a conserved lysine residue in myosin (K30) that gets ubiquitinated by UFD-2 is also targeted by the methyl-transferase SMYD1, with the K30 methylation improving myosin stability[48]. These data point to a competition of myosin modifiers to control myosin turnover in the cell.

Our data thus exemplifies how different components of the cellular PQC system collaborate and compete with each other to decide in concert about the fate of a given substrate protein, that is folding vs. degradation.

The stable complex of UNC-45:myo$^F$ with predominantly folded myosin implies that UNC-45 not only mediates the initial stages of protein folding, but also possesses a function beyond, a finding that is compatible with the role of UNC-45 in thick filament assembly and stress response[13,21]. Chaperone complexes, which engage with folded substrates, have also been observed in other biological pathways. For example, a specific J-domain protein was shown to bind natively folded tau in order to prevent protein aggregation[49]. Intriguingly, Tyr582 of the myosin FX$_3$HY motif has to fulfill two critical functions. First, Tyr582 has been shown to promote structural rearrangements in the myosin motor required for the ATP-driven power stroke[50,51]. In addition, as shown here, Tyr582 is one of three conserved myosin residues that is essential for UNC-45 binding. Given the importance of Tyr582 for motor activity and myosin folding, our findings suggest that binding of UNC-45 to myosin could have a regulatory role, coupling chaperone engagement with inhibition of ATPase activity. This regulation might be critical to keep the myosin motor in a resting state during protein folding, sarcomere assembly or under stress conditions. Consistently, a previous study highlighted the role of UNC-45 in inhibiting the myosin power stroke, and suggested that inhibition is achieved through interactions with the UCS domain[52]. The respective interplay between UNC-45 and myosin is now resolved in molecular detail, revealing a highly specific binding mode tailored to target a key functional motif of the client. Other client-specific chaperones are likely to also target conserved functional sites in their substrates, ensuring a high selective pressure to maintain specialized folding pathways.

Myosin-linked myopathies have been mostly studied in the light of deregulated myosin ATPase activity that impairs the actomyosin cross-bridge cycle[53] or the energy-saving super-relaxed state (SRX)[54,55]. Interestingly, mutations in chaperones like Bag3[56] and DnaJB6[57] cause congenital muscle defects, and HSPB6, HSP70 and HSP90 are upregulated upon cardiac failure[58,59], pointing to the importance of an efficient PQC system to prevent myosin-related myopathies. Consistently, human UNC45B mutations have been linked to mild forms of myopathy, attributed to reduced stability of the chaperone itself[17]. The characterized mutations are localized in the UCS domain, close to residues identified as myosin interaction sites (Suppl. Fig. 10), suggesting that they might impact not only UNC-45 stability, but also substrate interactions. We thus propose that myosin proteostasis, and specifically UNC-45:myosin interactions, play an important yet underestimated role in myopathies. The importance of a functional PQC system for preventing human myopathies is reflected by the Y582S mutation in the FX$_3$HY motif, underlying FSS developmental myopathy[28]. This point mutation abolishes the interaction with UNC-45, leading to myosin misfolding and aggregation. Our findings thus provide a direct link between impaired myosin quality control and the onset of myopathies, offering new paths for medical intervention. Moreover, our data indicate that disease causing mutations in myosin and UNC-45 are recapitulated by the *C. elegans* proteins. Given its powerful genetics and emerging role to study protein misfolded diseases[60–62], *C. elegans* should be an attractive model system to delineate cellular pathways causing myosin-based myopathies and explore counteractive strategies.

## Methods

### Protein production and purification from *E. coli*
SHE4 was amplified from *Kluyveromyces lactis* cDNA and cloned into a pET21a vector to generate the C-terminally His$_6$-tagged protein. The protein was expressed in BL21(DE3) cells which were induced with 100 µM IPTG and incubated at 18 °C overnight. Cells were harvested by centrifugation, resuspended in 50 mM Tris pH 7.2, 300 mM NaCl, 1 mM TCEP (Buffer A) and lysed by sonication. The cleared lysate was applied to a Ni-NTA Superflow column (Qiagen). The column was washed with buffer A and a high salt buffer containing 50 mM Tris pH 7.2, 1 M NaCl, 1 mM TCEP. SHE4-elution was performed with buffer A containing 60 mM imidazole. The elution fraction was loaded on a Bio-Gel HT Hydroxyapatite column (Bio-Rad) equilibrated with buffer A, and the protein was eluted in a linear gradient until 100 mM KH$_2$PO$_4$. For the final step of purification, SHE4 was subjected to size-exclusion chromatography using a Superdex 200 16/60 column (GE Healthcare) equilibrated with 10 mM Tris pH 7.2, 300 mM NaCl, 1 mM TCEP (for crystallization trials) respectively 10 mM HEPES pH 8.0, 150 mM NaCl, 1 mM TCEP (for ITC).

Full length *C. elegans* UNC-45 and UNC-45ΔUCS proteins were expressed and purified as previously described[13]. In short, expression was induced with 100 µM IPTG for 12 h at 18 °C. Cells were harvested by centrifugation and lysed by sonication in 50 mM NaH$_2$PO$_4$ pH 8.0, 300 mM NaCl (His-tag) or 100 mM Tris pH 8.0, 150 mM NaCl, 1 mM EDTA (Strep-tag). Tagged proteins were purified by affinity chromatography (HisTrap resp. StrepTrap, both GE Healthcare) with elution buffer containing 200 mM imidazole respectively 2.5 mM d-desthiobiotin. Proteins were then subjected to size-exclusion chromatography using a Superdex 200 16/60 column (GE Healthcare) equilibrated with 20 mM HEPES, pH 8.0, 150 mM NaCl. For ubiquitination assays UBA1 (*H. sapiens*), LET-70 and UFD-2 (both *C. elegans*) were purified as described previously[20].

Human UNC-45B was expressed from a PET-21a. In short, expression was induced with 100 µM IPTG for 12 h at 18 °C. Cells were harvested by centrifugation and lysed by sonication in 50 mM NaH$_2$PO$_4$ pH 8.0, 300 mM NaCl (His-tag). Tagged proteins were purified by affinity HisTrap with elution buffer containing 400 mM imidazole. Then, we performed SEC using a Superdex 200 16/60 column (GE Healthcare) equilibrated with 20 mM HEPES, pH 8.0, 150 mM NaCl.

### Protein production and purification from insect cells
The *C. elegans* myosin motor domain MHC (1–790) wildtype and mutants (F577A, H581A, Y582A, Y582F, Y582S, R580-GSGS-T585) were expressed as described previously[30]. MHC-B (*C. elegans*) containing a C-terminal His$_6$-tag was cloned into the polyhedrin promotor site of the pFastBac DUAL vector (Invitrogen). Baculoviruses for respective constructs were produced by transfecting Sf9-cells (Expression Systems) with FuGENE HD (Promega). For protein expression, High Five cells (Expression Systems) in suspension culture at a density of $1 \times 10^6$ cells/ml were infected with baculovirus (V1) and incubated at 21 °C for 96 h.

For myosin purification cell pellets were resuspended in 50 mM Tris pH 8.0, 300 mM NaCl (buffer A), 1:10 000 benzonase was added and cells were lysed with a homogenizer. The cleared lysate was loaded onto a NiNTA column (GE Healthcare) equilibrated with buffer A. The column was washed with buffer A, and myosin was eluted with buffer A, containing 200 mM imidazole. The elution fractions were applied to a Superdex 200 16/60 column (GE Healthcare) equilibrated with 20 mM HEPES pH 8.0, 150 mM NaCl. For purification of UNC-45-myosin complexes, the fractions eluting from the HisTrap column containing myosin were loaded on a Strep-Tactin gravity flow column (IBA Lifesciences) equilibrated with 100 mM Tris pH 8.0, 150 mM NaCl, 1 mM EDTA (Strep buffer A). The UNC-45-myosin complex was eluted with buffer A containing 2.5 mM d-desthiobiotin. To separate the different complex fractions, the sample was loaded onto a Superdex 200 3.2/300 column (GE Healthcare) equilibrated with 20 mM HEPES pH 8.0, 150 mM NaCl.

### Isothermal titration calorimetry (ITC)
For peptide quantification a tryptophan was introduced at the N-terminus. This residue is visible in the SHE4-myosin co-crystal structure but was omitted from the discussion since it is not part of the

native protein sequence. ITC for characterizing interactions between SHE4 and the synthesized myosin peptides was performed at 25 °C with a VP-ITC (Microcal). Peptides and protein were diluted in 10 mM HEPES pH 8.0, 150 mM NaCl, 1 mM TCEP. For wildtype SHE4, the temperature-controlled sample cell was loaded with 10 µM protein solution and titrated with 90 µM ligand in the mixing syringe (SHE4 N475A: 10 µM SHE4 in the cell, 107 µM myosin peptide in the syringe). An initial injection of 5 µl was followed by 29 consecutive injections of 10 µl with 300 s equilibration time in between. Control experiments with buffer and peptide were performed to correct for the heat of dilution. Data were analyzed with the MicroCal Origin software using a single side binding model. The peptides were synthesized in-house on a Syro Peptide Synthesizer (Multisyntech) and HPLC purified.

### Crystallization of SHE4 and SHE4-myosin peptide
SHE4-crystals were grown by sitting-drop vapour diffusion at 4 °C in 96-well plates. 100 nl protein solution (145.2 µM) was mixed with 100 nl reservoir solution (0.1 M HEPES pH 7.5, 10% PEG 8000, 20% ethylene glycol, 0.03 mM CaCl₂, 0.03 M MgCl₂) and yielded crystals after 3 days. The conditions used for co-crystallization of SHE4 with the myosin peptide were 0.1 M bicine/Trizma base pH 8.5, 10% PEG 20000, 20% PEG MME 550, 20 mM carboxylic acid.

The unit cell parameters of the P3₁21 space group were 129.5 Å, 129.5 Å, 77.2 Å (SHE4) respectively P3₁21 and 128.5 Å, 128.5 Å, 77.5 Å for SHE4 with the myosin peptide. Diffraction data of SHE4 were collected at the Deutsches Elektronen-Synchrotron (DESY) and for SHE-myosin peptide at the European Synchrotron Radiation Facility (ESRF). The crystal structure of SHE was determined by molecular replacement using the *C. elegans* UCS-domain as a search model (PDB: 4i2z). The initial model from Phaser[63] was submitted to rigid body and B-factor refinement in CNS[64] and yielded an improved electron density map. The initial model was built with O[65] and refinement of the final structure was performed with PHENIX. Ramachandran statistics were calculated with PHENIX. The apo-SHE4-structure was used for molecular replacement to help determine the crystal structure of SHE4 with the myosin peptide. For figure preparation PyMOL (Schrodinger) was used.

### Protein sequence analysis
For conservation analysis of the UCS-domain representative proteins from the UNC-45, CRO1 and SHE4 family were selected. For analysis of myosin conservation, all annotated myosin sequences from the Uni-Prot repository were used. Multiple sequence alignment was performed with MUSCLE[66] and conservation was plotted using Jalview[67] applying the BLOSUM62-score.

### Peptide spot assays
Myosin-peptides (MHC-B, *C. elegans*, residues 1–790) consisting of 15 amino acids were synthesized on a derivatized cellulose membrane (Intavis 32.100) using the ResPep SLi synthesizer (Intavis). The membrane was designed with an overlap of 12 amino acid residues compared to the neighbouring peptide. The membrane was activated with methanol, blocked with 3% bovine serum albumin (BSA) and incubated with 50 µg/ml Strep-tagged (C-term) wildtype UNC-45/UNC-45ΔUCS/UNC-45Δloop. The peptide spot membrane was blotted onto a nitro-cellulose membrane and UNC-45 binding was probed with an anti-Strep antibody (Qiagen) using an ECL-kit for detection (GE Healthcare).

### Thermal shift assays
These assays were performed in 96-well-plate format using a RT-PCR thermocycler (Biorad). 70 µl of wildtype UNC-45 or UNC-45 ΔUCS (1 µM), respective peptides (2 mM) and SYPRO Orange dye (5x) were prepared in 20 mM HEPES pH 8.0, 150 mM NaCl. For measurement, the plate was incubated at 12 °C for 3 min, then a gradient of 1 °C/min was applied up to 95 °C. Three samples for each combination of protein and peptide were measured.

### Fluorescence anisotropy
Peptides (sequence of *C..elegans* AHFAMRHYAG, of *H. sapiens* AHF-SLIHYAG, and mutations indicated in the figures) were synthesized in-house, as described in a previous section, and contained a N-terminal fluorescein. Fluorescence anisotropy experiments were performed in a buffer containing 25 mM Tris pH 8.0, 150 mM NaCl. A peptide concentration of 2 µM was used. The starting concentration of UNC-45 was 180 µM, (for UNC-45B 170 µM) which was subsequently diluted 1:1. Polarization was acquired with a PHERAstar microplate reader (BMG Labtech). Fitting was performed using a non-linear least squares algorithm in GraphPad Prism 8.

### Ubiquitination assays
Ubiquitination assays were performed in 20 mM Tris pH 7.5, 50 mM NaCl, 1 mM DTT, 10 mM MgCl₂. The reaction mix contained 20 µM DyLight800-Maleimide-ubiquitin, 0.5 µM E1 (UBA1), 1.2 µM E2 (LET-70), 1 µM E3 (UFD-2) and 10 mM ATP. To prepare the substrate for the assay containing in vitro reconstituted UNC-45:myosin-complex, equimolar ratios of both proteins were mixed and incubated at 4 °C or 27 °C for 60 min. 1.5 µM of this pre-incubated UNC-45:myosin was added to the ubiquitination assay reaction mix. When purified UNC-45:myosin-complexes were used as substrate, 1 µM of the indicated SEC-fractions were subjected to the ubiquitination assay after purification without prior incubation. Ubiquitination assays were performed at 21 °C, at the respective timepoints the reaction was stopped by adding SDS-PAGE loading buffer and heating the sample to 95 °C for 5 min. Samples were electrophoretically separated on BioRad Stain-Free TGX 4–20% gels. Proteins modified with ubiquitin were visualized by DyLight800-fluorescence and total protein levels were measured using Stain-Free tryptophan fluorescence, alternatively by Western Blot with a anti-His antibody (Qiagen). Signals were quantified using Fiji software.

### Analytical size-exclusion chromatography of UNC-45:myosin-complexes
SEC-fractions from UNC-45:myosin complex purification were split in half and incubated at 4 °C or 27 °C for 30 min. After incubation, samples were applied to a Superdex 200 3.2/300 column (GE Healthcare) equilibrated in 20 mM HEPES pH 8.0, 150 mM NaCl. 100 µl fractions were collected, proteins were analyzed by SDS-PAGE.

### Crosslinking mass spectrometry (XLMS)
For XL-MS, 0.5 mg/ml of the purified complex fractions or of in vitro reconstituted protein complexes was crosslinked in 20 mM HEPES pH 8.0, 150 mM NaCl using 0.25 mM DSS (Thermo Fisher) at 25 °C and 500 rpm for 30 min. The reaction was quenched with 50 mM ammonium bicarbonate at 25 °C for 10 min. Samples were evaporated to dryness at 45 °C (Concentrator, Eppendorf), resuspended in 8 M Urea and reduced with 2.5 mM TCEP. Subsequently, samples were alkylated with 5 mM iodoacetamide at room temperature in the dark for 30 min. For digestion, urea was diluted to 1 M by adding 50 mM ammonium bicarbonate and 2 µg Trypsin (Promega) per 100 µg protein. Samples were incubated at 37 °C for 20 h. Trypsin was inactivated by adding 0.4% (v/v) trifluoroacetic acid. For desalting, samples were loaded on Sep-Pak cartridges (Waters) equilibrated with 5% (v/v) acetonitrile, 0.1% (v/v) formic acid and eluted with 50% (v/v) acetonitrile, 0.1% (v/v) formic acid. To enrich for crosslinked peptides, the samples were separated by SEC. Peptides were resuspended in 30% (v/v) acetonitrile, 0.1% (v/v) trifluoroacetic acid and applied to a Superdex 30 3.2/300 column (GE Healthcare). Fractions containing crosslinked peptides were evaporated to dryness. For MS-analysis samples were resuspended in 5% (v/v) acetonitrile, 0.1% (v/v) TFA. The nano HPLC system used was an Ulti-Mate 3000 RSLC nano system (Thermo Fisher Scientific) coupled to an Orbitrap Fusion Lumos Tribrid mass spectrometer (Thermo Fisher Scientific), equipped with a Proxeon nanospray source (Thermo Fisher Scientific). Peptides were loaded onto a trap column at a flow rate of

25 µl/min using 0.1% TFA as mobile phase. After 10 min, the trap column was switched in line with the analytical column (both columns from Thermo Fisher Scientific). Peptides were eluted at a flow rate of 230 nl/min, and a binary 3 h gradient. The gradient started with 98% mobile phase A (water/formic acid, 99.9%/0.1%, v/v) and 2% mobile phase B (water/acetonitrile/formic acid, 19.92%/80%/0.08%, v/v/v), increased to 35% mobile phase B over 180 min, followed by a 5 min-gradient to 90% mobile phase B. Acquisition was performed in data-dependent mode with a 3 s cycle time. The full scan spectrum was recorded at a resolution of 60 000 in the range of 350–1500 m/z. Precursors with a charge state of +3 – +7 were fragmented. HCD-collision energy was set to 29%. The resolution of MS2-scans recorded in the Orbitrap was 45 000 with a precursor isolation width of 1.0 m/z. Dynamic exclusion was enabled with 30 s exclusion time.

Fragment spectra peak lists were generated from the raw MS-data using the software MSConvert[68] (v 3.0.20105) selecting the peak picking filter. Crosslink search was performed using XiSearch[69] (v 1.7.4) applying the following parameters: 6 ppm MS1-accuracy; 20 ppm MS2-accuracy; DSS-crosslinker with reaction specificity for lysine, serine, threonine, tyrosine and protein N-termini with a penalty of 0.2 (scale 0–1) assigned for serine, threonine and tyrosine; carbamidomethylation of cysteine as a fixed modification; oxidation of methionine as variable modification; tryptic digest with up to four missed cleavages; all other variables were used at default settings. Identified crosslinks were filtered to 5% FDR on link level with the software XiFDR[70] (v 1.4.3.1). For plotting of crosslink data, the in-house software CrossLinkingVisualizer was used[71].

### Alphafold2 modelling
The models were generated using a locally run version of ColabFold[72]. For the UFD-2 model, the full-length sequence of the *C. elegans* variant was used. For the *H. sapiens* UNC-45B/myosin model, either full-length UNC-45B or the UCS domain only (residues 452–931) were modelled with the minimal MYH3 peptide that is homologue to the one identified in the crystal structure (sequence: HFSLIHY).

### Haddock modelling
Cross-linking data were used as input are from the UNC45:myo$^F$ complex purification and analyzed with the Haddock pipeline[31]. The resulting clusters from the initial docking were used for refinement of the model. J-walk has been used to calculate the SASD between the links using the best cluster as input. We have defined crosslinks with a distance of >40 Å to be satisfied and used those links as input for HADDOCK to calculate a refined model.

### Solubility assay
Hi5 cells were infected for 4 days at 21 °C with a baculovirus expressing both UNC-45 WT and myosin variants in the same backbone, as for the "Protein expression" section. Cells were then spinned at 1000 G for 15 min, weighted and flash-frozen in liquid nitrogen. The pellets were then resuspended in 50 mM Na Phospate pH 8, 300 mM NaCl, 1:5000 benzonase. The volume of buffer for the resuspension was proportional to the weight of the pellet (5X). The suspension was centrifuged at 20000 G for 30 min to separate soluble and insoluble fractions. The supernatant of the clear cell lysate was then loaded on a 10% self-cast SDS-PAGE. Quantification of the band intensity was performed with ImageJ, and plotting and statistical analysis (paired *t*-test) was performed with GraphPad prism.

### Statistics and reproducibility
For the blots and SEC profiles in the paper, we provide here the number of repeats with similar results. Figure 1a, 6 independent replicates. Figure 2a, 2 independent replicates. Figure 2b, 2 independent replicates. Figure 5b, 2 independent replicates. Supplementary Fig. 2a, 2 independent replicates. Supplementary Fig. 2b, 2 independent replicates. Supplementary Fig. 3a, 2 independent replicates. Supplementary Fig. 3c, 2 independent replicates. Supplementary Fig. 9b, 2 independent replicates.

### Reporting summary
Further information on research design is available in the Nature Portfolio Reporting Summary linked to this article.

## Data availability
The mass spectrometry proteomics data have been deposited to the ProteomeXchange Consortium via the PRIDE partner repository with the dataset identifier PXD040428. Atomic coordinates have been deposited in the Protein Data Bank (PDB) with accession codes 8BRG (She4_apo) and 8BRH (She4_peptide). Source data are provided with this paper.

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

## Acknowledgements

We thank all members of the Clausen lab for critical remarks on the manuscript. Furthermore, we are grateful to the staff at DESY (Hamburg, Germany) and ESRF (Grenoble, France), where the diffraction data were recorded. We would also like to thank the VBCF Mass Spectrometry Facility in particular Elisabeth Roitinger and Karl Mechtler for their assistance as well as the VBCF Protein Technologies and Anita Lehner for helping with protein production. Moreover, we are grateful to Mathias Madalinski for synthesizing the peptides. AV was supported by the DOC fellowship from the Austrian Academy of Sciences (ÖAW), LD by a grant from the Vienna Science and Technology Fund (WWTF) GA No LS21-009, and TC by an Austrian Research Promotion Agency Headquarter grant 852936. The IMP is supported by Boehringer Ingelheim.

## Author contributions

Conceptualization, A.V., R.A. and T.C.; Methodology, A.V. and R.A.; Investigation A.V, R.A. and R.M.G.C.; Resources, D.S., L.D. and A.B.; Writing – original draft, A.V., R.A. and T.C.; Writing - Review and editing, R.A. and T.C.; Visualization, A.V. and R.A.; Supervision, A.M and T.C; Funding Acquisition, A.V. and T.C.

## Competing interests

The authors declare no competing interests.
