## [Transparent Peer Review file · Nature Communications]

UNC-45 assisted myosin folding depends on a conserved FX3HY motif implicated in Freeman Sheldon Syndrome

Corresponding Author: Dr Tim Clausen

Figures originally included in the author's rebuttal have been redacted from this file.

Version 0:

Reviewer comments:

Reviewer #1

(Remarks to the Author)

The authors build on their prior study investigating the interplay between the *C. elegans* chaperone UNC-45 and striated muscle myosin MHC-B. Here, the primary new findings are that UNC-45 forms distinct complexes with folded and unfolded myosin, UNC-45 binds to a specific conserved motif in the myosin heavy chain (FX3HY), and that a disease mutation in this motif impairs UNC-45 assisted folding. Cross-linking/mass spectrometry, size-exclusion chromatography and co-crystallization of a UCS domain with myosin peptides (spanning the FX3HY motif) are the main techniques used. They propose a model whereby unique interactions between UNC-45 and different folding states of myosin lead to divergent pathways (folding versus degradation) and outcomes for the myosin client.

The paper is well-written and interesting, and significantly adds to understanding the mechanism by which the myosin client interacts with its chaperone UNC-45. A few specific issues that should be addressed are listed below.

1. Please clarify why co-expression of UNC-45 with the MHC-B motor domain in insect cells leads to two species that have bound UNC-45 (Fig. 1a). I thought that the author's prior 2019 Nat Com paper showed that the myosin was fully folded and thus UNC-45 did not bind.
2. The rationale for using 27C should be stated.
3. Do the authors have any insight regarding why human cardiac and human striated muscle myosin when co-expressed with their cognate UNC-45 in insect cells do not result in folded myosin, leaving investigators to use the C2C12 muscle cell line for expression of recombinant mammalian striated myosins? What is different about striated myosin from *C. elegans*?

minor

1. Line 108 typo complexes

Reviewer #2

(Remarks to the Author)

This manuscript investigates the molecular mechanism of UNC-45, a chaperone that assists in the folding, assembly, and degradation of the muscle protein myosin. The authors demonstrate the molecular mechanism of UNC-45 upon myosin quality control using various techniques such as X-ray crystallography, crosslinking-mass spectrometry (XL-MS), and biophysical assays.

The authors identify that UNC-45 can form distinct complexes with folded (myoF) and unfolded (myoUF) conformations of myosin. HSP90 binds both complexes while UFD-2 preferentially binds and ubiquitinates myoUF destined for degradation. They show that unfolded Myosin makes a conserved FX3HY motif in the myosin motor domain available for UNC-45 binding. This was identified through peptide arrays and structurally validated by determining a co-crystal structure of a minimal UNC-45:myosin complex. The Y582S mutation linked to Freeman-Sheldon syndrome prevents this UNC-45 binding, causing myosin aggregation.

The manuscript is logically structured and clearly written overall. The key findings are adequately supported by the presented data.

Comments and concerns:

Regarding the tested temperatures of 4°C and 27°C, 27°C reflects physiological temperature, indicating most myosin will be unfolded under normal conditions. What is the biological rationale for comparing 4°C and 27°C?

The SEC profiles show an abundance of UNC-45:myoUF at physiological temperature. What explains this phenomenon?

You did not conduct experiments with human proteins. Humans have two UNC-45 homologs with low sequence identity to *C. elegans* UNC-45. Since myosin quality control seems tissue-specific, what evidence supports *C. elegans* as an appropriate model?

Did you predict the UNC-45:myo complex structure using AlphaLink2? This could provide insights into complex folding.

Line 105-109: 'Since the stoichiometry of UNC-45 and myosin was similar in all fractions, we hypothesized that the elution peaks reflect different conformational states of myosin bound to UNC-45. To test this, we compared the SEC elution profiles of UNC-45:myo complexes with those of folded and heat-denatured myosin alone.' I question whether oligomeric myosin states could exist in SEC peaks with similar UNC-45:myosin stoichiometry. Recommend confirming complex molecular weights using additional techniques like SEC-MALLS.

Line 294: 'Disruption of the FX3HY motif abolishes myosin folding in the cell.' The solubility assay has experimental variables (cell density, virus titer, etc.) that complicate data interpretation. Quantifying band intensities is also unreliable. Could you provide more robust functional data?

The Rfree value is relatively high considering the highest resolution of the dataset. Moreover, there is a planarity outlier that should be corrected. I would recommend further refinement to improve the Rfree, address the planarity outlier, and better fit the data at high resolution.

line 173 states that it is a functional complex. Does that refer to supplementary figure 3a?

In Fig 2e, statistical significance should be indicated for the increased ubiquitination activity observed on myoUF compared to myoF. Quantification should also be shown for 2 technical replicates.

Figure 1. e / Figure 3. How were the structural models generated? Furthermore, manually orienting proteins based solely on crosslinks could misrepresent true binding modes, as the lysine-reactive DSS crosslinker can only access residues with exposed sidechains. I am happy with the interpretation of the crosslinks due to all of the validation done in this manuscript but I will advise some improvements to these figures.

- Perhaps docking with Haddock or Dizviz, making use of the crosslinks as distance restraints, would allow for localization of a potential binding site.
- Alternatively predicting the complex with AlphaFold multimer and seeing if the crosslinks agree. Also, the region around the FX3HY motif could be predicted alone using the human proteins to see if AlphaFold also suggests the binding interfaces discovered in these other organisms.

Reviewer #3

(Remarks to the Author)

Reviewer #4

(Remarks to the Author)

This study investigates the molecular details of UNC-45 interacting with myosin, highlighting its central role in myosin quality control. The authors revealed by crosslinking coupled to MS analysis that UNC-45 forms distinct complexes with both folded and unfolded myosin, directing them to downstream chaperones and E3 ligases. Structural analysis of a minimal chaperone:substrate complex revealed that UNC-45 binds to a conserved FX3HY motif in the myosin motor domain. Disruption of this identified interface through mutagenesis hinders myosin maturation, leading to protein aggregation. Furthermore, the authors demonstrate that a mutation in the FX3HY motif associated with Freeman Sheldon Syndrome compromises UNC-45-assisted folding, resulting in a diminished functional myosin pool. These findings underscore the significance of a malfunctioning myosin quality control mechanism as a previously overlooked contributor to human myopathies.

In summary, this study represents a carefully performed biochemical, biophysical and structural analysis of UNC-45 activities towards different myosin conformations. It provides new significant mechanistic insight into the role of UNC-45 in myosin folding.

The main criticism relates to the lack of more in vivo data, such as experiments with *C. elegans*, which serves as an ideal model to validate the assumptions in a living organism, revealing in vivo defects and phenotypic changes in muscle tissue. I

would recommend, if possible, within a reasonable time frame, to include these experiments.

Author Rebuttal letter:

REVIEWER COMMENTS

Reviewer #1 (Remarks to the Author):

1. Please clarify why co-expression of UNC-45 with the MHC-B motor domain in insect cells leads to two species that have bound UNC-45 (Fig. 1a). I thought that the author's prior 2019 Nat Com paper showed that the myosin was fully folded and thus UNC-45 did not bind.

When co-expressed with UNC-45, the majority of myosin is part of the soluble fraction of the cell lysate, representing the folded, functional form of the motor protein (as shown in our previous work). In the present study, we have now performed a tandem affinity purification, using His-tagged UNC-45 and strep-tagged MHC-B, to selectively isolate myosin molecules bound to the cognate chaperone. As this fraction presents the minor pool of MHC-B in insect cells - accounting for about 1% of the total recombinant protein - the chaperone-bound form was overlooked in our previous analysis, also because the complex co-eluted with the high-molecular weight fraction of UNC-45 present in large excess. Thus, establishing a tailored purification procedure was instrumental to isolate the UNC-45:myosin complex and characterize two sub-complexes thereof, bound to distinct non-native myosin forms.

2. The rationale for using 27°C should be stated.

We have previously shown that in vitro, sample incubation at 27°C resulted in myosin unfolding, while UNC-45 remained stable. We therefore used this temperature to selectively unfold myosin in the myosin:UNC-45 complex, without affecting the chaperone. This rationale is stated in lines 120 fl.

3. Do the authors have any insight regarding why human cardiac and human striated muscle myosin when co-expressed with their cognate UNC-45 in insect cells do not result in folded myosin, leaving investigators to use the C2C12 muscle cell line for expression of recombinant mammalian striated myosins? What is different about striated myosin from *C. elegans*?

Our previous work showed that *C. elegans* UNC-45 can collaborate with the insect cell chaperone machinery to produce functional *C. elegans* MHC-B myosin. In contrast, it was so far not possible to express human cardiac or muscle myosin in functional form in insect cells, even not when co-expressing human UNC45B. We presume that UNC45B cannot team-up with the myosin folding factors in insect cells, likely because these are evolutionary too distant. We also attempted producing human muscle myosin in insect cells by co-expressing human UNC-45B, HSP70 and HSP90; however these trials have

1

not succeeded so far. Most likely, further (co)chaperones beyond HSP70/HSP90 and UNC-45 are required for myosin folding, such as specific substrate adaptors or regulatory proteins. A recent study from the Morimoto lab (Sui et al, 2022, bioRxiv) provided a concrete hint towards accessory chaperones. The authors show that small heat shock proteins of the HSP16 family are strongly upregulated in worms expressing a metastable myosin variant, suggesting that these chaperones are critical for keeping, or bringing, myosin in shape.

minor

4. Line 108 typo complexes

Thanks for noting. We corrected the typo.

Reviewer #2 (Remarks to the Author):

1. Regarding the tested temperatures of 4°C and 27°C, 27°C reflects physiological temperature, indicating most myosin will be unfolded under normal conditions. What is the biological rationale for comparing 4°C and 27°C?

With regards to *C. elegans* physiology, a temperature of 27°C presents a severe heat-shock, as worms are typically kept at 16°C, with stress responses being triggered at 21°C, and reducing lifespan. When applying heat-shock conditions in vitro, we saw that purified myosin gets destabilized and misfolds at 27°C. In contrast, the UNC-45 protein is stable at 27°C, providing a simple way of selectively destabilizing myosin within the myosin/UNC-45 complex. Given its relatively small protein stability, we conducted control experiments at 4°C, a temperature preserving the folded and functional state of myosin over prolonged times. Thus, our 4°C setting serves as an in vitro proxy for the biological non-stress condition of MHC-B myosin in *C. elegans* muscle cells.

2. The SEC profiles show an abundance of UNC-45:myoUF at physiological temperature. What explains this phenomenon?

Please note that the SEC profiles were obtained after the tandem affinity purification, selecting only for myosin molecules bound to UNC-45. We thus enrich for a rare species in which myosin is bound to UNC-45. Compared to the overall amount of myosin expressed in insect cells, this fraction is minimal, accounting for 1%.

2

3. You did not conduct experiments with human proteins. Humans have two UNC-45 homologs with low sequence identity to *C. elegans* UNC-45. Since myosin quality control seems tissue-specific, what evidence supports *C. elegans* as an appropriate model?

As indicated by the Reviewer, the quality control of muscle myosin is tissue specific, requiring specialized folding and degradation factors. Several lines of evidence support our notion that *C. elegans* UNC-45 is such dedicated factor, presenting an attractive model to study muscle myosin (mis)folding: (1) Similarly to human UNC45B, *C. elegans* UNC-45 is expressed only in muscle cells, reflecting its tissue and client specific PQC function. (2) The so far characterized chaperone network in muscle cells is highly conserved between human and *C. elegans* (Shemesh et al., Nature Comm, 2021). (3) Considering the high conservation of the identified FX3HY motif and its interacting residues in UNC-45, the targeting and substrate folding mechanism is likely conserved across eukaryotes.

To experimentally address this point, we cloned, expressed and purified human UNC45B. We then performed fluorescence anisotropy binding assays, testing the interaction with human myosin (MYH3) derived peptides containing the FX3HY motif and controls thereof. As for the *C. elegans* system, we observed a weaker binding for the Y582A/S mutant peptides. Our results (new Fig. 5e) confirm that the UNC-45:myosin binding mode is conserved from yeast to human, as is now mentioned in the main text. These data are further corroborated in silico, where we use AlphaFold2 to predict a minimal complex of human UNC45B and MYH3, aligning well to the presented chaperone:substrate co-crystal structure (Fig. 5d).

4. Did you predict the UNC-45:myo complex structure using AlphaLink2? This could provide insights into complex folding.

Following the reviewer's suggestion, we tried to predict the complex of UNC-45 and myosin with AlphaLink2, using the XLMS data. The resulting prediction, however, does not differ from the prediction from AlphaFold2, except for the positioning of the TPR domain (see appended figure). We presume that the difference in the crosslinker used for AlphaLink2 and the one used in our experiments might play a role. AlphaLink2 parameters were optimized for SDA (succinimidyl 4,4'-azipentanoate), that has a spacer arm of 3.9 Å, while in our experiments we used DSS (disuccinimidyl suberate), with a spacer length of 11.4 Å. We thus decided to use HADDOCK to predict the complex. In the resultant model, the β -hairpin harbouring the FX3HY motif is in close proximity to the myosin binding groove of the UNC-45 UCS domain, being more consistent to our findings than the AlphaFold2/AlphaLink2 model. We included these results in Supp. Fig. 2b.

[Redacted]

Modelling of UNC-45/myosin complexes (a) Superposition of AlphaFold2 and AlphaLink2 models of the UNC-45:myosin complexes. Both UNC-45 and myosin align well, with the exception of the TPR

domain that is present in different orientations. (b) Superposition of Alphalink-2 and Haddock models of the UNC-45:myosin complex.

4

5. Line 105-109: Since the stoichiometry of UNC-45 and myosin was similar in all fractions, we hypothesized that the elution peaks reflect different conformational states of myosin bound to UNC-45. To test this, we compared the SEC elution profiles of UNC-45:myo complexes with those of folded and heat-denatured myosin alone. I question whether oligomeric myosin states could exist in SEC peaks with similar UNC-45:myosin stoichiometry. Recommend confirming complex molecular weights using additional techniques like SEC-MALLS.

Following the suggestion of the reviewer, we performed mass photometry measurements of both peaks, and included the data in the new Supp. Fig. 1b. These data show that the UNC-45:myoUF complex has no discrete molecular weight, but rather spans a size range of 150-600 kDa. We presume that this sample contains a heterogenous mixture of chaperone, eventually present in tandems (Gazda et al, 2013), bound to misfolded and aggregated myosin molecules. Contrary, the UNC-45:myoF peak contains two discrete species: the UNC-45 chaperone alone (110 kDa) and UNC-45 in complex with one myosin molecule (200 kDa). As the mass photometry measurements had to be carried out at low protein concentration (<10 nM), the two species likely reflect the reversible formation of the chaperone:substrate complex.

6. Line 294: Disruption of the FX3HY motif abolishes myosin folding in the cell. The solubility assay has experimental variables (cell density, virus titer, etc.) that complicate data interpretation. Quantifying band intensities is also unreliable. Could you provide more robust functional data?

All experiments were performed upon carefully monitoring cell density, and upon co-expressing UNC-45 and myosin from the same vector backbone with the same promoter. Thus, levels of UNC-45 and myosin should be comparable in separate experiments. We agree that titre differences could cause effects on the solubility, with lower titres resulting in a better solubility due to a reduced burden to the insect cell chaperone machinery. To address this point, we infected insect cells with increasing amounts of a virus expressing UNC-45 and myosin, applied at 1:10 to 1:200 dilutions. When viruses were used at higher dilutions, we indeed observed slightly elevated protein levels of soluble wildtype and Y582A myosin. However, this minor effect does not account for the strong reduction of Y582A myosin in the soluble fraction of the cell lysate, as shown in the figures below. Moreover, in the original experiment, levels of expressed UNC-45 are very similar (<10% variations), suggesting comparable titres between different viruses. Having said this, we agree that our original statement has been too bold and changed the chapter title to "Disruption of the FX3HY motif hinders myosin folding in the cell".

[Redacted]

Titration of WT and Y582A mutant myosin in Hi5 cells. Left: SDS-Page of soluble fractions of Hi5 cells infected with different titres of a virus expressing UNC-45 and myosin. Right: Quantification of the myosin/UNC-45 ratio of the SDS-PAGE analysis.

7. The Rfree value is relatively high considering the highest resolution of the dataset. Moreover, there is a planarity outlier that should be corrected. I would recommend further refinement to improve the Rfree, address the planarity outlier, and better fit the data at high resolution.

We thank the reviewer for noting these inconsistencies. Following the advice, we further refined and improved the crystal structure, obtaining a structure with better stereochemistry and better R-free/R-work values, mostly by performing TLS refinement and correcting stereochemical outliers by hand (Table S1).

8. line 173 states that it is a functional complex. Does that refer to supplementary figure 3a?

The term "functional" was misleading. We removed it, as the statement solely refers to the reconstituted complex, formed at elevated temperature (Fig. 2b, right panel).

9. In Fig 2e, statistical significance should be indicated for the increased ubiquitination activity observed on myoUF compared to myoF. Quantification should also be shown for 2 technical replicates.

Considering that UFD-2 can ubiquitinate both UNC45 and myosin, and that the Dylight-ubiquitin experiment cannot discriminate between the two species due to their similar molecular weight, we decided to switch to a Western Blot analysis, monitoring myosin via its His-tag. Having purified four biological replicates of UNC-45:MyoUF and UNC-45:MyoF complexes, we performed ubiquitination assays with UFD-2. We quantified the intensities of ubiquitin bands by normalizing against the total

6

amount of myosin. For the UNC-45:MyoUF complex, we observed strong mono-ubiquitination aside poly-ubiquitination signals, whereas we did not detect any myosin linked ubiquitination when incubating UFD-2 with UNC-45:MyoF. These data support our notion that the UNC-45:MyoF complex contains mostly folded myosin that cannot be targeted and ubiquitinated by UFD-2. We updated the main figure Fig. 2e with data from the improved ubiquitination assay.

10. Figure 1. e / Figure 3. How were the structural models generated? Furthermore, manually orienting proteins based solely on crosslinks could misrepresent true binding modes, as the lysine-reactive DSS crosslinker can only access residues with exposed sidechains. I am happy with the interpretation of the crosslinks due to all of the validation done in this manuscript but I will advise some improvements to these figures. Perhaps docking with Haddock or Dizviz, making use of the crosslinks as distance restraints, would allow for localization of a potential binding site. Alternatively predicting the complex with AlphaFold multimer and seeing if the crosslinks agree. Also, the region around the FX3HY motif could be predicted alone using the human proteins to see if AlphaFold also suggests the binding interfaces discovered in these other organisms.

Following the reviewer suggestion, we used HADDOCK to dock myosin onto UNC-45 based on the XL-MS data. The best scoring prediction showed interaction between the U50 myosin domain which contains the FX3HY motif and the UCS domain of UNC-45. We included these results in Supp. Fig. 2b. We also tried to predict the complex of human UNC45B and MYH3 myosin with AlphaFold2. However, the resulting complex did not align with our XLMS data and the identified myosin FX3HY motif was not engaged in complex formation, most likely because the beta-hairpin is buried in the folded myosin (Supp. Fig. 6). Contrary, when we modelled a minimal complex comprising the UCS domain of UNC45B and the FX3HY peptide stretch, AlphaFold2 was able to predict a complex that closely mimics our crystal structure. These in silico data nicely corroborate our biochemical and structural findings and were included in Fig. 5d and Supp. Fig. 9.

Reviewer #4 (Remarks to the Author):

The main criticism relates to the lack of more in vivo data, such as experiments with *C. elegans*, which serves as an ideal model to validate the assumptions in a living organism, revealing in vivo defects and phenotypic changes in muscle tissue. I would recommend, if possible, within a reasonable time frame, to include these experiments.

We thank the reviewer for this remark. As indicated, obtaining in vivo data from a *C. elegans* myopathy model will be important; however, this is beyond the scope and timeframe of the present study that is

7

focused on the molecular mechanism of how UNC-45 channel myosin into folding and degradation pathways. We believe that the new in vitro and in silico data that extend our model to the human UNC45B:myosin complex strengthen the relevance of our study. We are confident that the presented findings will provide a solid framework to guide experiments addressing myosin PQC in vivo. In this context, a recent publication (Federica et al, 2024) revealed an interesting cross-connection to a further myosin chaperone, SMYD1. The authors show that SMYD1 methylate a specific lysine residue to enhance stability and assembly of muscle myosin in zebrafish. Notably, the same residue is targeted by the UFD-2 ubiquitin ligase, pointing to another layer of PQC regulation, as briefly referred to in our Discussion.

Version 1:

Reviewer comments:

Reviewer #2

(Remarks to the Author)

The authors have addressed my comments and concerns.

Reviewer #3

(Remarks to the Author)
